# Integration of Vestibular and Auditory Information in Ontogenesis

**DOI:** 10.3390/children9030401

**Published:** 2022-03-11

**Authors:** Elena I. Nikolaeva, Victoria L. Efimova, Eugeny G. Vergunov

**Affiliations:** 1Department of the Developmental Psychology and Family Pedagogics, Herzen State Pedagogical University, 191186 Saint-Petersburg, Russia; 2Prognoz Pediatric Neurological Clinic, Director, 191014 Saint-Petersburg, Russia; prefish@ya.ru; 3Laboratory of the Organism Functional Reserves, Federal State Budgetary Scientific Institution, Scientific Research Institute of Neurosciences and Medicine, 630099 Novosibirsk, Russia; vergounov@gmail.com

**Keywords:** paired integrative systems, vestibular system, auditory system, post-rotational nystagmus, auditory brain stem response, cervical vestibular evoked myogenic potentials, two-block partial least squares

## Abstract

Background: We analyzed the hypothesis that the vestibular and auditory systems are integrative functions. Methods: The study involved 383 children (5.5 ± 2.4 years old). We assessed the conduct of auditory information by recording the auditory brain stem response (ABR), post-rotational nystagmus (PRN), and cervical vestibular evoked myogenic potentials (cVEMP), and calculated the integration of the parameters. All procedures were carried out using the JACOBI 4 software package. Results: We have found out that PRN, ABR, and cVEMP represent three different groups of integrative functions, each of which is conditioned by its own integrative mechanism. We have proven that PRN and ABR are associated with age, but no relationship was found between cVEMP and age. Conclusion: According to our data, the severity of ABR and PRN depended on age, while cVEMP was not associated with age. The functional immaturity of the child’s vestibular system, which probably arose in utero, often becomes apparent only at school when reading and writing must be mastered. These skills require maturity of both the vestibule ocular and vestibule spinal functions of the vestibular system.

## 1. Introduction

The vestibular system is a unique sensory system that informs the body about its position relative to the Earth’s gravitational field. This sense is activated by the only external stimulus that never disappears as long as the organism is on Earth. While sunlight regularly disappears, and you can find places with eternal silence, and without odor or taste, there is always gravity on Earth. That is why a long stay of a person in space, where this constant human companion is absent, leads to a total change in all bodily systems [1,2].

The picture of the world, which is recreated by the brain, is established relative to the position of the person’s head. The brain receives data about the position of the head from the vestibular system. That is why, when turning the head and body, this picture does not change. Any disturbances in the vestibular system leads not only to somatic or autonomic problems, but also to a distortion of the picture of the world, and therefore cognitive problems. Our earlier studies revealed that all children from the first to the fourth grade, who have some kind of learning difficulties, had some kind of disturbance in the conduction of vestibular information to the brain. This is due both to the fact that during pregnancy the mothers of these children did not move much, and also to the fact that the children, after birth, led a sedentary lifestyle [3,4]. Thus, the psychological and somatic health of the child is associated with the quality of the formation of the vestibular system, and this, in turn, is determined by the way of life.

An important difference between the vestibular system and other systems is that it has no primary cortex location. It is represented in the cortex by a bilateral cortical network of neuronal populations in various areas. Processing of vestibular information occurs in the superior temporal gyrus, inferior parietal lobule, and the anterior insula of the left and right hemispheres [5,6].

The vestibular system is the first to appear both in evolution and in ontogeny [7]. It provides information that constitutes the mandatory background for any human activity, from determining the position of one’s own hands and feet relative to one’s head, to deciding what to do in a particular situation [8].

At the same time, the receptor part of the vestibular system does not represent a single unit, but consists of elements capable of reacting both to the gravitational field of the Earth itself and to the movement of body parts and the body itself in this field: i.e., both linear and angular acceleration. Moreover, this rather localized part of the vestibular system is represented by an unusually branched cortical network [8].

It is important to note that vestibular information is initially transmitted along the vestibulocochlear nerve; that is, both vestibular and auditory information are initially sent as part of different branches of the same nerve, and only after switching in the brain stem do their paths diverge. The first afferent projections of the vestibular nerve refer to the vestibular nuclei in the brainstem and cerebellum. The vestibular nuclei play an important role in the formation of the motor reflexes that control eye movements and posture. In particular, the vestibule–ocular reflex [9] ensures that eye movements correspond to head movements: when the head is turned, the eyes move in the opposite direction, which helps stabilize the image on the retina.

The vestibule spinal reflex (VSR) [10] maintains a person’s posture while head movement is compensated for by changes in muscle tone. Thus, at the level of the vestibular nuclei, separate somatosensory, proprioceptive, and visual inputs of information are formed. In general, these low-level multisensory control pathways play an important role in stabilizing the spatial relationship between the body and the environment during movement [11].

The central part of vestibular processing is the Parieto-Insular Vestibular Cortex, which includes the operculum, the parietal cortex, the secondary somatosensory cortex, the inferior parietal cortex, the superior temporal cortex, the posterior insula, and the premotor cortex [12,13]. For this study, it was important that the vestibular network is bilateral: even with unilateral excitation of this network, the response is obtained from both the left and right sides. In this case, the central representation of the vestibular system is ipsilateral; that is, there is dominance of the ipsilateral hemisphere in relation to the stimulated ear [14,15] and handedness [16].

In addition, vestibular signals are projected into areas traditionally labeled as related to other functions, such as visual, somatosensory, motor, memory, or affective [17].

Olson and Miller [18] put forward the hypothesis that the body has integrative systems—that is, a subset of morphological characters that change together during ontogenesis and evolution. The division into subsets is carried out either formally, according to correlations between features (R-groups), or substantively, according to the principle of joint functioning of features (F-groups). Ideally, both divisions should match.

Most researchers admit that the concept of integration is applicable at various biological levels [19,20], including genetic and ecological integration [21,22], and integration of the right and left sides of an organism’s body [23,24,25,26,27].

All of the above allowed us to assume the presence of an integrative system, specifically, the conjugate formation of the vestibular and auditory systems in ontogenesis. The purpose of this work was to identify evidence of the presence or absence of such conjugation in the ontogeny of the auditory and vestibular systems. The presence or absence of systems integration will make it possible to more accurately create recommendations for children with problems associated with both the function of the vestibular and auditory systems.

## 2. Materials and Methods

The study involved 383 children from 1.1 to 13.7 years old (mean age 5.5 years, standard deviation 2.4 years).

### 2.1. Assessment of the Conduct of Auditory Information

The functioning of the brainstem and the rate of transmission of auditory information by the structures of the brainstem were studied by recording the auditory brain stem response (ABR). The ABR rate allows us to evaluate the quality of myelination of individual auditory information conduction sites, which predetermines the child’s ability to perceive not only the auditory information, but also the speech information [28]. The first ABR represents postsynaptic activities of the first auditory neurons. ABR II and III are primarily generated within the pons, with possible contributions from the auditory nerve; ABR V in the midbrain (inferior colliculus); ABR IV and VI originate from the pons and the medial geniculate body, respectively [29,30]. ABR was measured by the NikoletVikingselect-TM analyzer (VIASYS Healthcare, Inc., San Diego, CA, USA). A modified technique called the VI peak method [31] was used. To measure the VI peak, we used a modified stimulus: a short tone burst with a filling frequency of 4000 Hz, a plateau duration of 0.5 ms, and a leading edge of 0.5 ms, with an intensity of 70 dB above the hearing threshold.

The use of such a modified stimulus made it possible to set the time of conduction of an auditory signal along the brain stem (from the hair cells of the organ of Corti to the medial geniculate body of the thalamus). The identification of the VI peak was carried out, taking into account the detection of I, III, and V peaks with standard stimulation. Averages calculated from 500 to 1000 presentations on the left and right sides were made [32].

We used leads on the mastoid process on the left and the vertex on the right. Silver chloride cup electrodes were fixed using an adhesive conductive paste, and a ground electrode was placed at the Fpz point. The interelectrode resistance did not exceed 4 kΩ. We summarized 500–1000 evoked responses (each with a duration of 12 ms) without traces containing artifacts (with their automatic rejection when the amplitude discrimination threshold of 30–40 mV was exceeded). The bandwidth of the signals was set in the range from 100 to 3000 Hz. On the analyzed track with modified stimulation on the presentation side, the dominant V peak was determined. The next positive deviation was considered the VI peak, and its peak latency was determined.

The stimuli were presented through headsets (TDH39) separately to the left and right ears at a frequency of 10.1 Hz.

During the study, the child sat in a comfortable chair. Preschool children sat on the lap of one of the parents.

### 2.2. Post-Rotational Nystagmus with the Registration of the Electrooculogram

Post-rotational nystagmus assessment is a well-described, validated procedure in the literature [33,34]. The function of the semicircular canals of the vestibular apparatus was assessed by recording the post-rotational nystagmus (PRN) with the Rehacor-T psychophysiological telemetric system devised by the MEDICOM-MTD Company (Taganrog, Russia) in Encephalan-MPA software system.

The subject was placed in a sitting position in a Barany chair. His or her head was set in an inclined forward position at an angle of 30 degrees. The horizontal component of the electrooculogram (EOG) was recorded using two EOG leads; the electrodes were located at the outer corners of the eyes, and the neutral electrode was in the center of the forehead. The chair rotated manually at a speed of 10 revolutions in 20 s. After the end of the chair rotation, the PRN was recorded until its complete attenuation; the control was carried out on the monitor screen. The duration of PRN was assessed; on the basis of preliminary studies, 12 s was considered the norm. Dysfunction was considered the duration of the PRN being less than 12 s. A difference between the duration of nystagmus after rotation of the chair to the right and to the left of more than 30% indicated an asymmetry in the reactivity of the semicircular canals.

### 2.3. Cervical Vestibular Evoked Myogenic Potentials (cVEMP)

The otolith function of the vestibular apparatus was assessed using the cervical vestibular evoked myogenic potentials (cVEMP) method [35,36,37]. cVEMP in response to sound stimulation was recorded on a neuro-averaging device called the “Neuro-MEP-4” (“Neurosoft”, Ivanovo, Russia). The latency of the P13 cVEMP wave was recorded from m. Sternocleidomastoideus on the side where the clicks were presented (sacculo-cervical reflex). Acoustic stimulation was used to vibrate the otolith receptors (a short sound stimulus with an intensity of 120 dB with a duration of 0.5 ms was presented through headphones). We averaged 5–20 cVEMP in 10 superposition runs to assess the reproducibility of responses.

The subject was placed in the chair in a sitting position; the head was maximally retracted to the shoulder, which provided the necessary tension in the m. Sternocleidomastoideus. The electrodes were positioned as follows: the negative electrode was fixed in the area of the lateral edge of the upper sternum at the point of attachment of m. Sternocleidomastoideus; the positive one was fixed in the upper part of it on the side of stimulation; the ground electrode was placed in the center of the forehead. The latency period of P13 was evaluated; preliminary studies have shown that 10 ms is the norm for children under 15 years of age.

### 2.4. The Assessment of the Integration of the Parameters

Mantel’s test is an instrument of multivariate analysis, which allows estimation of the correlation (parametric or rank) between the distance matrices between the studied variables [38,39]. In our case, such variables are the series of psychophysiological data (including those traits that consist of gradations 0 and 1) in subgroups of subjects. The result of the Mantel’s test is the coefficient r_m_, which varies from 0 to +1, and shows the closeness of the correlation relationship (without the direction of this relationship) [40].

PLS analysis is a method of obtaining Projection to Latent Structure, the original name of which is Partial Least Squares. Two-Block PLS analysis (2B-PLS) is an effective tool of PLS analysis [41]. 2B-PLS analysis in the case of psychophysiological data allows for revealing deep independent “latent structures” which act as psychophysiological mechanisms simultaneously for two different blocks (matrices B1 and B2) of multidimensional indicators [42].

During the 2B-PLS analysis, both blocks are centered, scaled, and rotated to obtain the maximum covariance between the score matrices (B1-score and B2-score), which are the projections of matrices B1 and B2 on the latent structures sought. The resulting latent structures are described using orthogonal loadings matrices (B1-loadings and B2-loadings).

The rows in matrices B1 and B2 are data on subjects, and the columns are indicators. Thus, the indicators act as initial coordinate axes (including those correlated with each other), and can be regarded as “explicit structures”, each of which accounts for some (usually small) amount of total variance. The goal of 2B-PLS analysis is to find a system of pairs of axes for both blocks at once that expresses the maximum covariance pattern [43]. In this case, load matrices are transition matrices from initial explicit structures to find new latent structures.

As a result of 2B-PLS analysis, there is an effective accumulation of all information from initial data series (the number of which can reach many hundreds) into several first independent latent structures. All PLS tools allow for situations when the number of initial indicators is more (and even many more) than the number of subjects.

All procedures (Data Preprocessing & Processing) were carried out using the JACOBI 4 software package [42].

## 3. Results

### 3.1. Data Preprocessing

To test the hypothesis about the integration of the paired traits under study (that is, about their joint change in ontogeny), the correlation between their indicators was first calculated from the sample (Table 1), and then the correlation with age.

Since for our sample the threshold of significance of the correlation coefficients was 0.1267 (*p* < 0.05 with α = 0.05 and Power = 1 − β = 0.80), the correlations (Table 1) are significant. Thus, the studied paired features (candidates for integrated features) can be attributed to integration functions of the R-type (formally, there is a correlation between paired indicators) and to functions of the F-type (meaningfully, there is a joint functioning of features).

After that, by means of gradations of variables from 0 to 1, we brought them to a single scale and shifted their centers to a place common for all indicators: these operations do not violate mutual relations in the space of the studied indicators.

Mantel’s test showed that between the Euclidean distance matrices between the six indicators (Table 1) for boys and those for girls, the correlation coefficient is r_m_ = 0.9963 (R^2^ = 0.99). This allows us to consider our sample, including boys and girls, to be homogeneous with respect to the indicators we are studying.

Since we study ontogenesis, the first years of life are of particular scientific interest (we recorded the age of children to the nearest 0.01 year). In this case, we expectedly obtained a Lognormal distribution. Goodness of fit (Anderson-Darling (AD) & Kolmogorov-Smirnov (KS)) showed:-for Normal Distribution P_KS_ << 0.01 (Reject Normal) and P_AD_ = 5.7 × 10^18^ (Reject Normal), -for Lognormal Distribution P_KS_ > 0.15 (Can’t reject Lognormal) and P_AD_ = 0.090 (Can’t reject Lognormal).

This distribution (Figure 1) is due to the fact that there is a physical restriction on the left, associated with the date of birth of the child, and no physical restriction on the right.

In connection with the above, our study refers to the type of correlational pseudolongitudes (a one-step vertical slice across several cohorts of ages).

The distribution of raw data for the studied indicators is given in Figure 2, Figure 3 and Figure 4.

Then the subjects were grouped into separate matrices, for which rotations were performed in pairs (this does not violate mutual relationships and provides the maximum covariance between both score matrices) and load matrices were formed in the course of Two-Block PLS analysis.

### 3.2. Data Processing

The composition of the blocks of variables is shown in Table 2. Since our task was to rotate the variables describing the indicators (ABR, PRN, cVEMP) by the angle that gives the maximum covariance with age, only one variable (age) was included in block #2. Therefore, after Two-Block PLS analysis, we received only one Latent Structure, which is age (Figure 5).

Each pair of traits is a separate group, and only ABR (oppositely directed) and PRN (codirectional) indicators are associated with age (Figure 5, Table 3).

Thus, we have shown that ABR and PRN are the integration functions. The cVEMP score, while showing a correlation of mean strength between the left and right variables, does not show an association with age.

## 4. Discussion

We have already noted that vestibular and auditory information is transmitted to the nervous system within the framework of a single entity, the vestibulocochlear nerve, but this information only partially constitutes an integrated system. For each type of information, the integration of left and right indicators was found, the development of which goes hand in hand.

Previously, when examining 30 right-handed and 30 left-handed subjects (The laterality handedness quotient was assessed with the 10-item inventory of the Edinburgh test, [42]), Kirsch et al. [43] tried to identify the handedness-dependent organizational patterns of functional subunits within the human vestibular cortex areas. These categories revealed a familiar handedness-dependent bilateral vestibular network of multiple asymmetric and symmetric multisensory areas organized around a core region in the parieto-insular cortex, i.e., the middle, posterior, and inferior insula. This core region was hemispherically lateralized and multisensory in nature. Its surrounding areas were hemispherically symmetric, well-connected, and functionally influenced by the neighboring sensory systems.

It is considered that these interhemispheric balanced (symmetric) regions serve as integrative hubs to a core region in the parieto-insular cortex [44].

Unlike the peripheral processing of vestibular information, the cortical processing of vestibular information is asymmetric [45].

The asymmetry of network lateralization and handedness-dependency displayed in the work by Kirsch and her coauthors might reflect the dominance of the right hemisphere in right-handers for the vestibular information and the dominance of left hemisphere for speech.

We have shown that parameters ABR and PRN depended on age, while cVEMP did not depend on it. This reflected the multifunctional organization of the vestibular system, which was formed in the process of evolution as a result of the emergence of new tasks facing the body [46]. At the same time, the regulation of the position of the body in space ends before the age that was included in the analysis (one year), since it is of high importance in the process of giving birth to a child. The child’s regulation of eye position changes along with head movement is likely to be extensively corrected after birth, when visual stimulation appears. It is possible that the vestibular system lateralized in the process of evolution before the motor system and pre-determined the lateralization of phylogenetically later-developing cognitive functions [47].

Age-dependent ABR is also associated with the widespread incorporation of sound information into various processes, including the development of speech, which occurs after birth.

The significance of our results is determined by the possibility of predicting the adjustment of each of the three indicators in children in the event of certain changes in development caused by intrauterine problems and/or problems that arise in the child during childbirth or immediately after birth. If a function continues its development after ontogenesis, then the probability of it being corrected increases sharply. However, if the critical period of the development of the function has ended by the time of birth, then the correction of the function is either impossible or extremely difficult [48].

It has been shown that dysfunctions of the otolith part of the vestibular system are more often detected in school-age children with learning difficulties [49]. This means that the inability to process information about gravity and linear head movements is not compensated for by age without special training. The impulse from the vestibular nuclei in the brainstem is carried out in two directions: to the muscles and to the cerebral cortex. Earlier [48], we found that in children with learning difficulties, distortion of vestibular information has already occurred at the stage of registration. The functional immaturity of the child’s vestibular system, which probably arose in utero, often becomes apparent only at school when reading and writing must be mastered. These skills require maturity of both the vestibule ocular and vestibule spinal functions of the vestibular system.

The movement of the mother during pregnancy leads to the rocking of the baby in the amniotic cavity and the stimulation of the vestibular receptors. This promotes the development of the vestibular system, which generally ends before the baby is born. It is very important to tell expectant mothers about the importance of their physical activity for the future health of the child.

Both reading and writing require knowledge of the exact position of the head, hand, and sheet of paper relative to the hand. Since studying at school requires not only mastering written speech, but also perceiving large amounts of information by ear, children who have not formed the basic levels of integration of vestibular and auditory information are unsuccessful in learning and gaining a high level of intelligence development. The information we have obtained about the role of the vestibular system in teaching a child to read and write complements the understanding of the psychophysiological mechanisms of dyslexia, which may not be a permanent state, but the result of an unfinished process of functional maturation of the vestibular system in ontogenesis.

Considering our data in the general context of child development, we would like to emphasize that they fully correspond to the idea of asynchrony of maturation processes of different brain systems [49]. At the same time, our data show that the part of the vestibular system related to vestibule spinal function completes its development before birth. It is of great importance in the future cognitive development of the child. The disorders that develop in utero can only be detected when the child is in school, but they are very difficult to correct. Many school problems can be prevented by an early (before the age of one year) assessment of the child’s vestibular system.

## 5. Conclusions

Of all the indicators we studied, three groups of paired integrative functions were identified and described: right and left indicators for PRN, ABR (6R and 6L), and cVEMP (P13R and P13L). We have proven that PRN and ABR are associated with age, but no relationship was found between cVEMP and age.

We also found that PRN, ABR, and cVEMP represent three different groups of integrative functions, each of which is conditioned by its own integrative mechanism.

### Limitations

In the standard apparatus for evaluating the ABR, a click is used as stimulus. This exposure is effective for obtaining the first five peaks, but obtaining the sixth peak in this case is probabilistic. In our study, the sixth peak was induced not by click, but by tone, because the apparatus we used allows it. However, not all manufacturers of apparatuses for evaluating ABA include the option of changing stimuli. That is why there are not enough studies on children, especially for children under two years of age, with data on the sixth peak of ABR. Our study needs to be continued in order to get the number of subjects of each age.

## Figures and Tables

**Figure 1 children-09-00401-f001:**
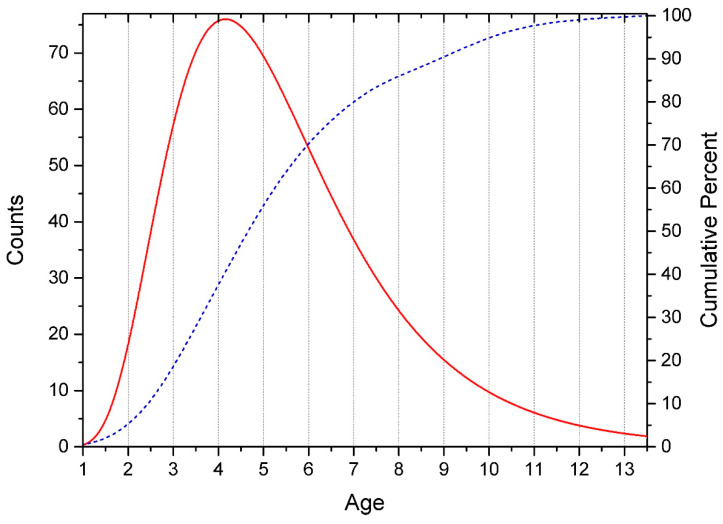
Age distribution of subjects. The solid line is the number of subjects by year of life (histogram approximation). Dotted line—cumulative percentage in the sample.

**Figure 2 children-09-00401-f002:**
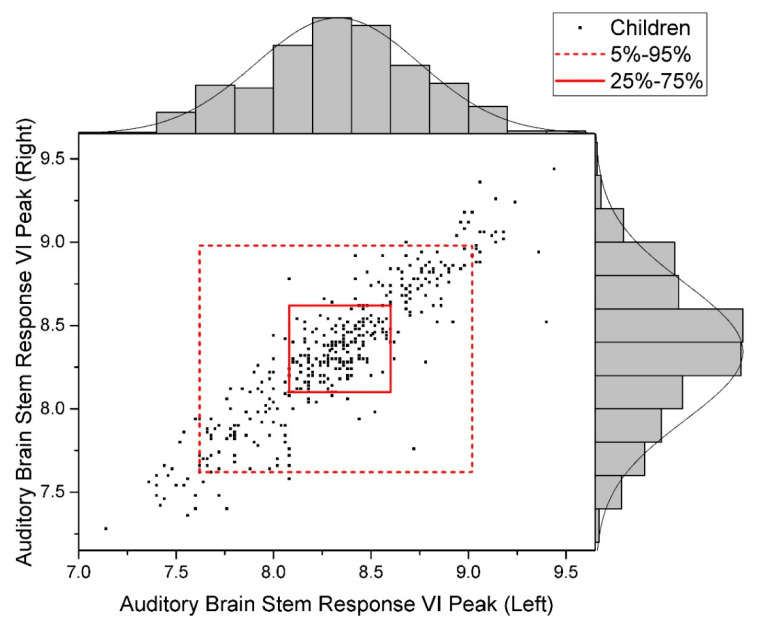
Data Distribution of Auditory Brain Stem Response, VI Peak (Normal).

**Figure 3 children-09-00401-f003:**
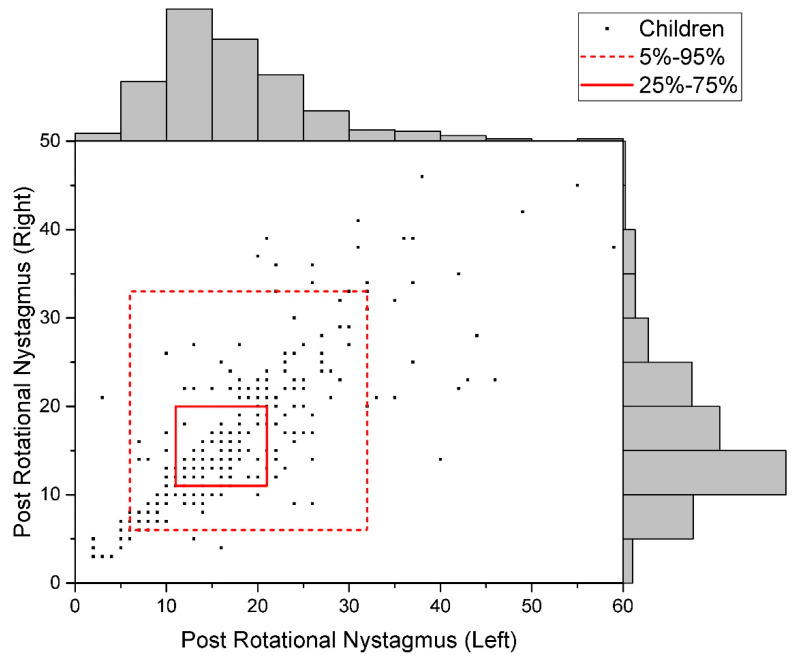
Data Distribution of Post-Rotational Nystagmus (Uniform).

**Figure 4 children-09-00401-f004:**
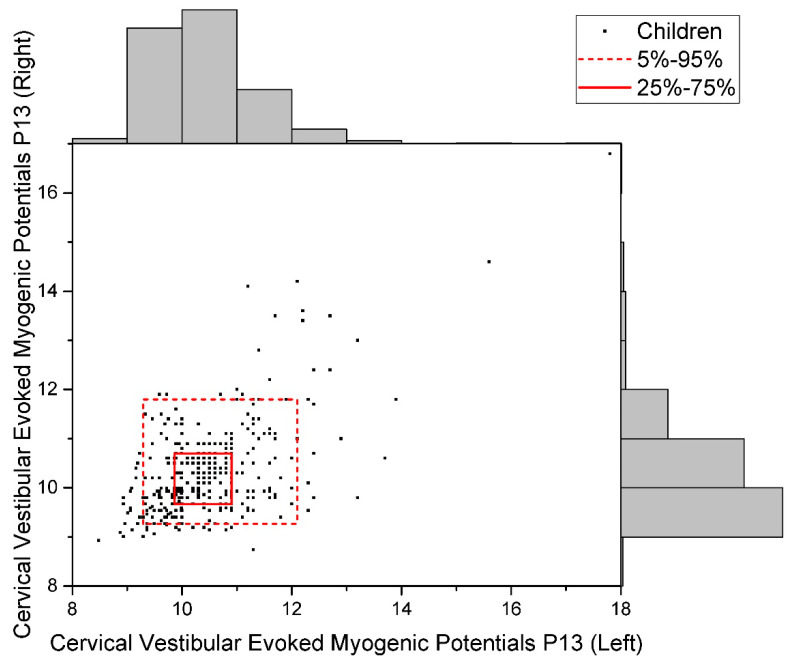
Data Distribution of Cervical Vestibular Evoked Myogenic Potentials P13 (Uniform).

**Figure 5 children-09-00401-f005:**
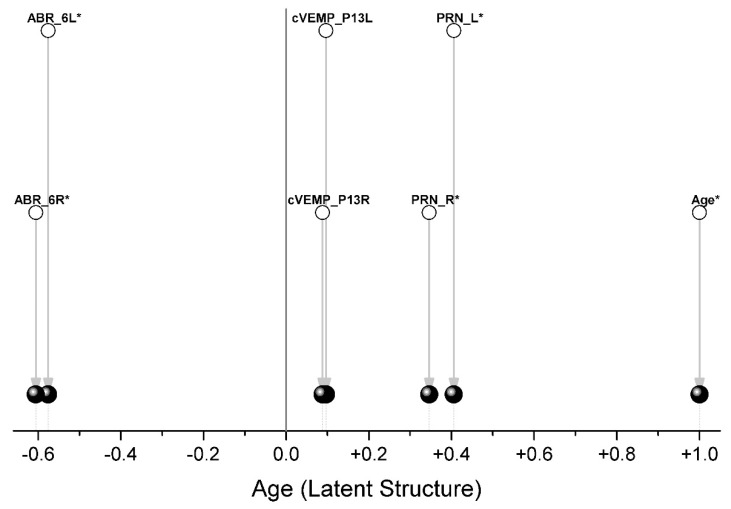
Loadings from Two-Block PLS analysis for variables from Table 2. ** the* correlation *coefficient is significant at the p < 0.05 level; the differences between the correlation coefficients themselves are shown in Table 3*.

**Table 1 children-09-00401-t001:** Pearson correlation between indicators of integrative functions.

Integrative Functions	Left vs. Right
Auditory Brain Stem Response, VI Peak	r = +0.907; R^2^ = 0.82
Post-Rotational Nystagmus	r = +0.803; R^2^ = 0.64
Cervical Vestibular Evoked Myogenic Potentials, P13	r = +0.573; R^2^ = 0.33

Notes: all correlation coefficients are significant at the *p* < 0.05 level.

**Table 2 children-09-00401-t002:** Blocks of variables in Two-Block PLS analysis (Age vs. Tests).

Code	Variable	Block
ABR_6L	Auditory Brain Stem Response, VI Peak, Left	#1
ABR_6R	Auditory Brain Stem Response, VI Peak, Right	#1
PRN_L	Post-Rotational Nystagmus, Left	#1
PRN_R	Post-Rotational Nystagmus, Right	#1
cVEMP_P13L	Cervical Vestibular Evoked Myogenic Potentials, P13, Left	#1
cVEMP_P13R	Cervical Vestibular Evoked Myogenic Potentials, P13, Right	#1
Age	Age	#2

**Table 3 children-09-00401-t003:** Differences between loadings (correlation coefficients) for the studied indicators after Two-Block PLS analysis.

Indicators	Left vs. Right	PRN	cVEMP_P13
Auditory Brain Stem Response, VI Peak	not found	*p* < 0.05	*p* < 0.05
Post-Rotational Nystagmus	not found		*p* < 0.05
Cervical Vestibular Evoked Myogenic Potentials, P13	not found		

## Data Availability

The research was conducted on the basis of the Herzen State Pedagogical University (Saint-Petersburg) with the participation of the Prognosis clinics, the materials are stored in the department of the developmental psychology and family pedagogics, access to them may be granted by permission of the head of the department (e-mail: klemtina@yandex.ru).

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
