# Peer review of "Integration of Vestibular and Auditory Information in Ontogenesis"

_children, 2022, doi:10.3390/children9030401_

Round 1

Reviewer 1 Report

In this manuscript presented by Nikolaeva et al., the authors working hypothesis was that there is an integration of vestibular and auditory functions during development. To test the hypothesis, the authors measured ABR, PRN and cVEMP in ~400 children under 14 years old. The introduction section seems adequate, but the rest of the manuscript falls short grossly. Not much detail is provided on the actual analysis done to test the hypothesis. The results presented are meager to make any meaningful interpretation or to comment on. In this reviewer’s opinion, this manuscript reads like a first draft.

Author Response

In this manuscript presented by Nikolaeva et al., the authors working hypothesis was that there is an integration of vestibular and auditory functions during development. To test the hypothesis, the authors measured ABR, PRN and cVEMP in ~400 children under 14 years old. The introduction section seems adequate, but the rest of the manuscript falls short grossly. Not much detail is provided on the actual analysis done to test the hypothesis. The results presented are meager to make any meaningful interpretation or to comment on. In this reviewer’s opinion, this manuscript reads like a first draft.

We have added information on the specifics of conducting barrel information with links to English-language sources. We expanded the results by presenting them in figures and commenting on them

Reviewer 2 Report

The article deals with the presence or absence of integration of vestibular and auditory information in ontogenesis, which is very important to better understand the neural network of these systems and, thus, improve the early detection of interventional changes, in order to contribute to the good -being of the child development.

The manuscript has a clear objective, but the methodology needs to be improved to respond adequately.

The experimental design must be better described so that the results can be adequately evaluated. Example: the method used to evaluate the ABR - VI peak method - is not adequately described and the article used as a reference was not found in the databases and archives of the journal itself. It also looks like it´s only written in Russian. Thus, the method can hardly be reproduced by other researchers.

Information about the population studied that the results was not inclued. Only the age variable was used to discuss the integration of auditory and vestibular information. The limitations of the study were not pointed out.

The data need to be further discussed, especially on the biological plausibility of ontogenesis in relation to hemispheric lateralization and age - the main findings

Author Response

The manuscript has a clear objective, but the methodology needs to be improved to respond adequately.

The experimental design must be better described so that the results can be adequately evaluated. Example: the method used to evaluate the ABR - VI peak method - is not adequately described and the article used as a reference was not found in the databases and archives of the journal itself. It also looks like it´s only written in Russian. Thus, the method can hardly be reproduced by other researchers.

Information about the population studied that the results was not inclued. Only the age variable was used to discuss the integration of auditory and vestibular information. The limitations of the study were not pointed out.

The data need to be further discussed, especially on the biological plausibility of ontogenesis in relation to hemispheric lateralization and age - the main findings

We have added information about English-language sources regarding obtaining the ABR and expanded the description of obtaining ABR peaks . We changed the description of the results and added figures to specify the results. We added a Limitation section.

Round 2

Reviewer 1 Report

In the revised manuscript, the authors have provided additional details on methodology. The results section is also improved with additional figures. This revised version is much improved from the original. With that said, lines 102 to 104 are directly copied from the abstract of reference #30. These need to be paraphrased. 

Author Response

In the revised manuscript, the authors have provided additional details on methodology. The results section is also improved with additional figures. This revised version is much improved from the original. With that said, lines 102 to 104 are directly copied from the abstract of reference #30. These need to be paraphrased. 

answer:

we've replaced the repeats: 

According to our data, the severity of ABR and PRN depended on age, while cVEMP was not associated with age.

According to our data, the severity of ABR and PRN depended on age, while cVEMP was not associated with age.

To: 

According to our data, the severity of ABR and PRN de-pended on age, while cVEMP was not associated with age.

We have shown that parameters ABR and PRN depended on age, while cVEMP did not depend on it.

Reviewer 2 Report

For the study to be carried out, the evaluation method of the era must be previously validated

Author Response

For the study to be carried out, the evaluation method of the era must be previously validated

answer:

We do not use the term "era" in our work; it is not in the text of the article. Correspondingly, there are no methods for estimating "era".
All of the methods we use in this paper have been validated, which is reflected in the publications of their developers and researchers from various countries who apply them. The references to such works are given below.For methods 2 and 3 the works were added to the text of the article.

  1. Assessment of the conduct of auditory information, auditory brain stem response (ABR)

[28] Wan, G., Corfas, G. Transient auditory nerve demyelination as a new mechanism for hidden hearing loss. Nat Commun. 2017 Feb 17, 8, 14487. doi: 10.1038/ncomms14487.PMID: 28211470

[29] Hashimoto, I., Ishiyama, Y., Yoshimoto, T., Nemoto, S. Brain-stem auditory-evoked potentials recorded directly from human brain-stem and thalamus. Brain. 1981, 104(Pt 4), 841-859. doi: 10.1093/brain/104.4.841. PMID: 6976818 DOI: 10.1093/brain/104.4.841

[30] Hashimoto, I. Auditory evoked potentials recorded directly from the human VIIIth nerve and brain stem: origins of their fast and slow components. Electroencephalogr Clin Neurophysiol Suppl. 1982, 36, 305-314. PMID: 6962027

[31] [28] Efimov, O.I.; Efimova, V.L.; Rozhkov, V.P. Violation of the speed of the auditory information in the structures of the brain stem in children with speech disorders and learning difficulties. Sensory systems 2014, 28(3), 36–44. (In Russian)

[32] Chiappa, K. H. Evoked potentials in clinical medicine. NY: Raven Press, 1990. XVII, 647 p

  1. Post rotational nystagmus with the registration of the electrooculogram

Lotfi, Y., Rezazadeh, N., Moossavi, A., Haghgoo, H.A., Rostami, R., Bakhshi, E., Badfar, F., Moghadam, S.F., Sadeghi-Firoozabadi, V., Khodabandelou, Y. Rotational and Collic Vestibular-Evoked Myogenic Potential Testing in Normal Developing Children and Children With Combined Attention Deficit/Hyperactivity Disorder. Ear and hearing 2017, 38 (6), e352–e358. DOI: 10.1097/AUD.0000000000000451

Rine, R.M., Wiener-Vacher, S. Evaluation and treatment of vestibular dysfunction in children. NeuroRehabilitation 2013, 32 (3), 507–518. DOI: 10.3233/nre-130873

  1. Cervical vestibular evoked myogenic potentials (cVEMP)

Zhou, G., Dargie, J., Dornan B., Whittemore K. Clinical uses of cervical vestibular-evoked myogenic potential testing in pediatric patients. Medicine 2014, 93(4), e37. DOI: 10.1097/MD.0000000000000037.

Young, Y.-H., Chen, C.-N., Hsieh, W.-S., Wang S.-J. Development of vestibular evoked myogenic potentials in early life. European journal of paediatric neurology 2009, 13 (3), 235–239. DOI: 10.1016/j.ejpn.2008.04.008.

Wiener-Vacher, S. R., Quarez, J., Le Priol A. Epidemiology of vestibular impairments in a pediatric population. Seminars in hearing 2018, 39(3), 229–242. DOI: 10.1055/s-0038-1666815.

  1. Mantel's test, instrument of multivariate analysis

[33] Mantel, N., Valand, R.S. A technique of nonparametric multivariate analysis. Biometrics 1970, 26, 547–558.

[34] Shiryaev, A.G., Ravkin, Yu.S., Efimov, V.M., Bogomolova, I.N., Tzibulin, C.M. Spatial typological differentiation of the biota of claviod fungi of Northern Eurasia. Siberian Journal of Ecology 2016, 5, 648-660. DOI: 10.15372/SEJ20160503.

[35] Polunin, D.; Shtaiger, I.; Efimov, V. JACOBI4 software for multivariate analysis of biological data. bioRxiv, 2019. https://doi.org/10.1101/803684.

  1. 2B-PLS (Two-Block Projection to Latent Structure, Two-Block Partial Least Squares), instrument of multivariate analysis

[35] Polunin, D.; Shtaiger, I.; Efimov, V. JACOBI4 software for multivariate analysis of biological data. bioRxiv, 2019. https://doi.org/10.1101/803684.

[36] Rohlf, F. J., Corti, M. (2000). The use of two-block partial least-squares to study covariation in shape. Systematic Biology, 49(4), 740–753. doi: 10.1080/106351500750049806

[37] Kovaleva, V.Yu.; Pozdnyakov, A.A.; Litvinov, Yu.N.; Efimov, V.M. Assessment of the conjugation of morphogenetic and molecular genetic moduli of variation in the common vole Microtus s.l. in gradient environmental conditions. Ecological genetics 2019, 17(2), 21–34, https://doi.org/10.17816/ecogen17221-34.

We thank the reviewer for his in-depth analysis of our work, which greatly improved it
